# Double Jeopardy in Contemporary China: Intersecting the Socioeconomic Gradient and Geographic Context on Early Childhood Development

**DOI:** 10.3390/ijerph17144937

**Published:** 2020-07-08

**Authors:** Wangyang Li, Minyi Li, Yongai Jin, Shiqi Wang, Yi Zhang

**Affiliations:** 1School of Sociology, Beijing Normal University, Beijing 100875, China; wangyangli@pku.edu.cn; 2Faculty of Education, Beijing Normal University, Beijing 100875, China; shiqiwang@mail.bnu.edu.cn (S.W.); 201611010171@mail.bnu.edu.cn (Y.Z.); 3School of Sociology and Population Studies, Renmin University of China, Beijing 100872, China; jinyongai0416@ruc.edu.cn

**Keywords:** early developmental milestones, geographic context, socioeconomic gradient

## Abstract

Family socioeconomic status (SES) differences in early childhood development (ECD) are well documented, as are the neighborhood effects in early development outcomes. However, little is known about whether the SES gradient in ECD outcomes varies across geographic contexts by county-level variables in contemporary China. This study examines the effects of county-level socioeconomic background on inequalities in the developmental outcomes of young Chinese children. Individual-level child development data based on four early development milestones—taking a first step, first sentences, counting 10 objects, fully independent toileting—were combined with family- and county-level socioeconomic data from the China Family Panel Studies (CFPS). Using a hierarchical linear model (HLM) to examine how the broader socioeconomic context plays a role in the attainment of developmental milestones at expected times as young children grow and develop, we have found significant cross-level interaction effects between family SES and county-level variables in relation to developmental milestone attainment. The family SES gradient in the achievement of children’s developmental milestones is steeper for those in the under-developed regions than their counterparts in the more developed regions. Our findings suggest that low-SES children who are living in socioeconomically deprived regions suffer from a double disadvantage in terms of early development outcomes. Further research would be needed to contextualize the observed interactions and better explain the underlying mechanisms.

## 1. Introduction

The early childhood period, particularly from the prenatal to the first 5 years of life, has long been recognized to lay the foundations for a lifetime of well-being. An extensive body of scientific evidence suggests that early adversity and poor development have sustainable consequences on physical and emotional health outcomes later from a life course perspective, such as chronic diseases [1,2,3], psychiatric disorders [4], and depression [5]. Developmental delays in early childhood have also been shown to hinder brain development and lead to poorer educational attainment and economic performance in adulthood [6,7,8,9,10]. In a thorough review, Richter et al. [11] summarized the benefits of investing in high-quality early childhood education and care in improving health and well-being across the lifespan, particularly for disadvantaged children.

A large volume of research has well established the potential importance of macro-level contexts, mainly as related to geographic socioeconomic conditions, for the health and well-being in the first years of a child’s life [12,13]. Even after controlling for individual and family characteristics, children growing up in socioeconomically deprived neighborhoods are more likely to exhibit significant deficits than their counterparts from more affluent neighborhoods, such as delayed language and cognitive skills [14,15], and mental and physical health problems [16,17]. Efforts to explain why geographic residence affects child development have produced three fundamental mechanisms linking residence with child development [18]. The first mechanism emphasizes institutional resources, arguing that neighborhood economic status determines the availability, quality, and affordability of resources and opportunities that may alter child development. Leventhal and Dupéré [12], for example, in a very comprehensive review, showed that neighborhood disadvantage is negatively associated with children’s outcomes because of a lack of access to public and private services, such as child care and school, in such neighborhoods. The second mechanism highlights parental relationships, such as parental attributes, parenting practices, and the quality of the home environment. It maintains that low-socioeconomic status (SES) parents living in an impoverished neighborhood are more likely to experience psychological distress due to economic hardship and, consequently, lead to more harsh parenting and poorer home environment that can hamper children to achieve their developmental potentials [19,20,21,22]. Also, neighborhood poverty could potentially limit parents’ access to social support networks and subsequent child outcomes [20,23,24]. The last mechanism posits that neighborhood influences operate through the extent of formal and informal institutions that are present to monitor a resident’s activities and the presence of physical hazards (i.e., crime and violence). Decades of evidence suggest that neighborhood disorder and exposure to violence have a detrimental impact on children’s development and growth, and partially explain the association between neighborhood disadvantage and children’s outcomes [13].

In addition, researchers have increasingly explored how the macro environment interacts with micro-level factors to shape child development at the individual level. At the micro-level, family SES has often been found to be a significant determinant of child cognitive function [25,26], motor skills [27], and social-emotional development [21]. It is argued that children from low-SES families are always at a disadvantage, primary because of poverty, nutritional deficiencies, inadequate cognitive stimulation, and poor environmental conditions [28]. These SES gradients have been documented in many parts of the world [29,30,31]. There are two competing explanations to explain how and why these SES gradients in early childhood development (ECD) outcomes vary across geographic contexts. The first explanation argues that low-SES children who are living in severely deprived areas are facing double disadvantage in health. There is some evidence that children from low-SES families are especially at higher risk of developmental delays in deprived areas, while ECD outcomes are less sensitive to family SES in the more affluent ones, because child care resources and services are much more constrained by a mother’s knowledge and socioeconomic situation in those poor areas [32,33,34,35,36]. Moreover, a majority of families tended to move to more advantaged neighborhoods, which definitely benefits children [37]. That is, improved socioeconomic contexts may lessen or eliminate the association between family SES and child development. In contrast, the second explanation claims that disadvantaged children fare worse if living in relatively advantaged areas than if living in more deprived areas because they might feel they are being deprived in comparison with neighbors with the higher socioeconomic position [38,39,40]. In other words, higher community SES significantly initiates and/or enlarge the effects of family SES on child development. Robert [41] referred to these patterns as a double jeopardy hypothesis and a relative deprivation hypothesis, respectively. However, prior studies have not reached a consensus as to how these influences operate developmentally.

To date, nearly all studies on this subject have used data from the U.S. and Western European countries, and very few studies have been carried out to examine this question in East Asian countries such as China [42]. Many studies on China mainly focus on exploring the individual socioeconomic determinants of ECD and have yet to examine the geographic heterogeneity, and, most importantly, the interaction between individual and contextual socioeconomic conditions. The paucity of studies in China partly results from the absence of data combining information at the individual, family, and community levels. It is these mechanisms that are the focal point of analysis of the contributions of this paper.

To revisit this question is particularly important in China because geographic disparities in today’s China are tremendous. As studies have documented, social and economic inequality in China is heavily driven by structural or institutional factors such as residence and *hukou* (household registration system), in contrast to the Western societies, where individual and family characteristics have the most profound influences [43,44,45]. In their most recent study, Liu et al. (2020) observed a strong association between where Chinese children live and their cognitive function and high-school enrollment [46]. Despite substantial progress over the last decades, developmental delays in infancy remain a noticeable problem in China, particularly amongst children living in most deprived areas. As estimated, 17 million children younger than 5 years were at high risk for developmental delays and extreme poverty in 2010 [47].

In this study, we capitalize on data from the China Family Panel Studies (CFPS), a recently available national representative and longitudinal survey dataset, and investigate the effects of geographic socioeconomic context on inequalities in ECD in contemporary China. We aim to address the following research questions: (1) Do macro-level socioeconomic circumstances are associated with ECD in China? (2) How do macro-level variations in the socioeconomic context shape the relationship between individual-level socioeconomic position and ECD? This study improves our knowledge about whether and how inequalities in early childhood health and well-being differ across geographic contexts in East Asian societies.

## 2. Materials and Methods

### 2.1. Sample

This study drew on data from the CFPS, an on-going, nationally representative, and longitudinal survey launched in 2010 by the Institute of Social Science Survey (ISSS) at Peking University [48]. In the baseline survey conducted in 2010, the CFPS has successfully interviewed 14,960 households in 25 provinces, along with 33,600 adults and 8990 children within these households. The individuals in the baseline survey as well as core new family members residing in the target family have been tracked in the four waves of follow-up (2012, 2014, 2016, and 2018). In our study, we include information from the five waves. For those who were interviewed twice or more, we used their most current information. For example, for a child who was interviewed in 2010, 2012, and 2016, we used his/her information from CFPS 2016 rather than that in previous years. A critical feature of the CFPS is that it collects information at the individual, family, and community level, which allows us to link children with their families and communities to examine our research questions. The adult and child questionnaires are answered by individual adults and children age 10 or older, while information for children under age 10 is collected by a questionnaire that their guardians, mostly parents, answer. The CFPS was approved by the ethics committee of the Peking University (Ref No. IRB00001052-14010).

### 2.2. Measures

#### 2.2.1. Outcome Variables

As an outcome of health and development status in the early years, we used four developmental milestones in infancy, including motor, language, cognitive skills, and self-help. Although children do not develop skills on a rigorous timetable, there is a normal range and mean age in which a child may reach each milestone [49]. The timing of milestone achievement is viewed as an important indicator of child development [50], because later achievement has been shown to be linked to cognitive impairment [51,52,53], behavior problems [54], and psychiatric disorders [55,56,57] in adolescence and beyond.

Even though some studies have found children attain developmental milestones at substantially different ages across sexes and cultures [58,59,60,61], updated research has shown that the median age of attainment was equivalent for 96% milestones across sexes and 76% milestones across societies in terms of seven domains of child development: expressive and receptive language, gross and fine motor, play, relating, and self-help [50]. Our dataset did not involve cross-country variables, standardized development screening and assessment tools, which might avoid the influence of cultural norms. Furthermore, our selections of the development milestones are based on some practical reasons. In CFPS, a parent or adult guardian was asked to retrospectively report a child’s age (in months since birth) of walking alone, speaking in brief sentences, counting 10 objects, and fully independent toileting, respectively (summarized in Appendix A
Table A2). Previous studies have demonstrated the reliability of parent-reported age of achieved milestones to measure child development [62,63]. The CFPS has incorporated several measures, such as hard and soft range checks, to enhance accuracy of self-report data [64]. In this study, the composite variables were created using data from the five waves of CFPS and, therefore, captured the children′s skills and developmental progress. To ensure our results are not sensitive to this parametric specification, we created a series of dummy variables indicating whether being in the upper quantile for the four continuous responses (walking ≥16 months, speech ≥25 months, counting to ten ≥41 months, and toilet training ≥37 months). In other words, the age of milestone achievement of >75% is indicative of delays in the affected area of development.

#### 2.2.2. Independent Variables

Individual-level variables: Family SES is generally measured by parent’s education, occupation, and family’s income. Our study is based upon retrospective data, in which a parent or adult guardian reported child development and socioeconomic status. Hence, in this study, we are unable to use family income and parent’s occupation for the year of interview to measure the family’s economic resources. Instead, we used maternal education as a proxy of family SES, because it is (1) broadly stable throughout the adult life course, (2) highly correlated with other SES measures, such as occupation and income, and (3) most frequently used to measure SES in studies of child development [65]. For education, we used the self-reported years of schooling from the adult questionnaire. The results shown in Appendix A
Figure A1 suggest a strong correlation between maternal education and family income per capita for the most recent interview year. To corroborate our findings, we also entered family income per capita as the independent variable in our robust analyses, shown in Appendix A
Figure A2.

In addition, a series of pre- and postnatal factors have been identified to be associated with the timing and attainment of early developmental milestones [66,67,68,69]. We, therefore, control a set of covariates, such as a child’s gender, minority status, birth cohort, birth weight, the period of gestation, and mother′s age at delivery. Of particular importance in contemporary China is the *hukou* system, a divide between rural and urban residents. The rural areas are characterized by limited access to many resources and opportunities, such as education and health care [70]. We thus control for *hukou* before age 3 in this study, one for urban and zero for rural.

County-level variables: As we discussed earlier, social life is deeply influenced by socio-economic disparities across geographic regions in contemporary China. In this study, county-level characteristics were constructed by matching family addresses at the time the respondent child was 3 years old or below to the census geocode. We then used gross domestic product (GDP) per capita and the percentage of urban population of the county in which children resided before age 3 to measure geographic socioeconomic conditions.

The detailed results are shown in the Appendix A
Table A1.

### 2.3. Statistical Analysis

We estimated two-level hierarchical models for the four outcomes, respectively (walking alone, first sentences, counting 10 objects, and fully independent toileting), to explore the implications of contextual factors for ECD by examining the effects of geographic socioeconomic conditions on the timing of reaching early developmental milestones. Model 1 is the unconditional means model. For the *i*th person in the *j*th county, the model is
(1)$$ECD_{ij}=\pi_{0j}+\varepsilon_{ij}$$
(2)$$\pi_{0j}=\beta_{00}+\gamma_{0j}$$
where $\varepsilon_{ij}$~N(0, σ^2^) and $\gamma_{0j}$~N(0, τ^2^). The likelihood ratio test shows whether the two-level hierarchical model fits the data well. Model 2 adds county-level GDP per capita (logged) and the percentage of urban population in total population to examine whether children living in more developed regions demonstrate earlier attainment of milestones. Model 3 and Model 4 add further individual-level variables. Model 3 is the random intercept model, whereas Model 4 is the random coefficient model, that is, the one in which the coefficient of maternal education is set as random at the county level. Model 5, the full model, augments Model 4 with the cross-level interaction between maternal education and county GDP. In terms of regression equations, we have the equation as follows:(3)$$ECD_{ij}=\pi_{0j}+\pi_{1j}edu_{ij}+\pi_{2j}\vec{X_{ij}}+\varepsilon_{ij}$$
(4a)$$\pi_{0j}=\beta_{00}+\beta_{01}GDP_{j}+\beta_{02}Urban_{j}+\gamma_{0j}$$
(4b)$$\pi_{1j}=\beta_{10}+\beta_{11}GDP_{j}+\beta_{12}Urban_{j}+\gamma_{1j}$$
(4c)$$\pi_{2}=\beta_{2}$$

The model treats the intercept and education as random, and the effects of the control variables as fixed. The error terms of Equations (4a) and (4b) are assumed to be multivariate normally distributed, each with a mean of zero, nonzero variances, and zero covariances. Combining Equations (3), and (4a)–(4c), we obtain the following function in Equation (5):(5)$$ECD_{ij}=\beta_{00}+\beta_{10}edu_{ij}+\beta_{2}\vec{X_{ij}}+\beta_{01}GDP_{j}+\beta_{02}Urban_{j}+\beta_{11}edu_{ij}GDP_{j}+r_{0j}+\gamma_{1j}edu_{ij}+\varepsilon_{ij}.$$

All the statistical analyses were performed using STATA 16.0 (Stata Corp LLC, College Station, TX, USA).

## 3. Results

### 3.1. Descriptive Statistics

Table 1 reports descriptive statistics. The first row shows that the mean age of achieving developmental milestones in our sample was 14 months for walking without assistance, 20 months for first sentences, 34 months for counting 10 objects, and 32 months for fully independent toileting separately. We also present the percentile distribution of age with respect to the developmental milestones in Appendix A
Table A3. For example, a majority of children take the first step on their own at an age of between 10 months and 24 months in our analyses. In other words, there is considerable variance.

The attainment of developmental milestones (i.e., the starting age of achieving developmental milestones and percent reporting developmental delays) by explanatory variables is given in Table 1 and a correlation matrix of variables in our analyses in Appendix A
Table A4. Most of these correlations have the expected signs and highly significant as well. Obviously, the age reaching developmental milestones in infancy is steadily decreasing with maternal education. For example, the starting age of counting 10 objects is 26 months for children with a college-educated mother and 40 months for children whose mother has no education. Accordingly, the lower maternal education, the higher risks of children with developmental delays in motor, language, cognitive, and self-help skills. Results also indicate that later attainment of developmental milestones is negatively associated with a child’s gender, birth cohort, minority status, birth weight, gestational age, and *hukou* status.

### 3.2. Multi-Level Analyses

This section illustrates how individual- and contextual-level variables correlate with a variety of developmental milestones in the early years. To begin with, the first five columns report the association between motor development and the main predictors. In Model 1, the likelihood ratio test shows that the grouping variable at the county level is significantly associated with the mean age of motor milestone acquisition. This justifies the introduction of the county-level variables into the models. In other words, multi-level modeling should be employed instead of some usual form of regression.

Model 2 adds the county-level GDP per capita (logged) and urban population as percentage of total. The variance component representing variation across geographic regions decreases significantly from 1.063 to 0.706, which is about 34% of the total variation and can be explained by county-level socioeconomic conditions. As presented in Table 2, growing up in a county with a higher level of GDP per capita significantly reduces the starting age of successful walking (*p* < 0.001). More precisely, every 1000 yuan of increase in county-level GDP per capita is associated with a 4.016-month reduction of children’s age starting to walk without assistance. Results also show children residing in a county with a higher percentage of urban population are more likely to walker earlier, though the coefficient fails to reach statistical significance. As hypothesized, socioeconomic conditions at the county level constitute a powerful predictor of early motor development.

We further add individual variables in Model 3, namely, the random intercept model. Consistent with previous findings in different countries, Chinese children with high-educated mothers are more likely to walk without assistance at an earlier age. In Model 3, a one-year increase in maternal education is significantly associated with a 0.071-month reduction in the age of walking alone. Besides, net of individual-level factors, county GDP and percent of urban population are still negatively associated with later attainment of the motor milestones. More important for our purpose, however, is the fact that the significant effects of affluent regions on the achievement of early developmental milestones persist even after controlling for individual and family characteristics.

Model 4 is the random coefficient model including both individual- and county-level variables, which is to test whether the effects of maternal education on early motor development vary by geographic socioeconomic conditions. The likelihood ratio test shows that the random coefficient model fits the data better than the random intercept model. Accordingly, the estimated coefficient of maternal education varies significantly according to place.

Based on Models 5, while both maternal education and county GDP show significantly negative effects on the starting age of walking, the coefficient for the interaction term is significantly positive, indicating that residing in impoverished counties have particularly detrimental effects for low-SES children. That is, in regions with a lower level of social and economic development, children with low-educated mothers are more likely to walk later than their counterparts whose mothers are well-educated. In contrast, maternal education matters less in more developed regions. Hence, the relationship between maternal education and early development milestones is, to a large extent, mediated by geographic variations. A converging gap between the most- and least-educated groups is observed across regions.

Besides, we find similar patterns in the attainment of language and cognitive skills in the early years. After adjusting for all individual- and contextual-level variables simultaneously, there is a gradual reduction in the starting age of children speaking in sentences, counting 10 objects as well as fully independent toileting with the economic growth of a country where children grow up. Moreover, as demonstrated by the significantly positive effects of the interaction terms, poor and disadvantaged children are more likely to experience delays in achieving developmental competence, in particular for those living in less developed counties, resulting in greater socioeconomic disparities in health in such counties.

To better interpret the results, we graphically showed the changing associations between maternal education and the age of attainment of developmental milestones by county-level socioeconomic conditions, i.e., mean of GDP per capita minus one standard error, mean, and mean plus one standard error. As shown in Figure 1, low-educated mother′s flatter slopes, together with larger intercepts, of the regression line, suggested that education inequalities in the age of passing developmental milestones indeed change across regions, being greater in less developed areas than in developed ones.

To corroborate our findings, we add an interaction term between maternal education and county-level percentage of urban population to assess the geographic socioeconomic conditions on the achievement of early developmental milestones. The results are presented in Appendix A
Table A5 indicate that the positive coefficients of the interaction term support the decrement to these neighborhood effects associated with low-SES families. Also, we used a subsample of children under the age of 5 when interviewed since a short recall period may minimize recall bias. The models yield comparable results, shown in Appendix A
Figure A3.

To sum, significant geographic variations in ECD are present in contemporary China. On average, children are more likely to take a first step, speak in brief sentences, be able to count up to 10, and toilet independently earlier in socioeconomically advantaged areas. Additionally, although maternal education can facilitate children’s learning and growth, geographic socioeconomic conditions moderate the relationship between maternal education and ECD and reduce the inequality generated by maternal education.

### 3.3. Sensitivity Analyses

We use alternative coding schemes for early developmental indicators to do robustness checks. As aforementioned, the attainment of developmental milestones can also be measured on a dichotomous variable. We employed multi-level logistic regression models to check the robustness of our results, as shown in Table 3. Results from logistic regression and linear regression models are mostly consistent. Allowing the interaction between maternal education and county-level GDP per capita provides good evidence of different SES gradients in ECD across geographic locations. Highly educated mothers have substantially lower rates of children manifesting developmental delays, and the education effect is more pronounced for children growing up in the economically underdeveloped regions.

## 4. Discussion

Recent studies have focused on disparities from early on through macro-micro lens, particularly the correlation between where people live and their opportunity to quality-and length-of life. For example, Chetty and his colleagues [71,72] have found that early experiences and disparities in one’s life, from the quality of health service and preschool teacher to the neighborhood one grew up in, can have lasting effects. Their studies reveal that neighborhoods affect intergenerational mobility primarily through childhood exposure. However, what this issue looks like is still under research in contemporary China, especially tackling early childhood. Additionally, there has been little population-based research into the timing and attainment of developmental milestones in China. In this study, we used the age of reaching early developmental milestones, i.e., motor development, language and cognitive development, and self-help skills, as indicators of ECD. Capitalizing on recent data from the CFPS, we employed multi-level models to account for the contextual effects of geographic socioeconomic conditions on the attainment of developmental milestones in the early years.

Our analyses have shown that, in the context of contemporary China, there is a significant association with ECD and local socioeconomic conditions. Generally speaking, children in counties with lower per capita GDP and percentage of urban population in total population experience higher risks of delays in reaching developmental milestones than those from counties with higher per capita GDP and percentage of urban population, even after controlling for individual- and family-level variables. The finding is consistent with previous studies in other parts of the world [15,73,74]. As summarized by the two most comprehensive reviews, living in areas of social disadvantage is associated with a lack of access to institutional resources, low parenting quality, and poor environmental conditions, which in turn bear on child development [12,13]. In contemporary China, significant inequality in the geographic distribution of health resources is still evident, despite a more equitable per capita distribution of resources since the new health-care reform in 2009 [75]. Recent statistics also reveal increasing health inequality across geographic regions with rapid economic growth [76]. For example, in 2018, the health expenditure per capita was ¥ 2275 in Beijing with the highest GDP, roughly two times that in Gansu Province, where the GDP is the lowest [77]. It is suggested that, on average, children living in less socioeconomically developed regions are more likely to be exposed to developmental risks such as poverty with limited access to interventions than their counterparts from highly socioeconomically developed ones. Such regional differences place children from more developed regions at a great advantage from early on.

In addition, we find that geographic socioeconomic conditions contribute to attenuate SES gradients in ECD. It is evident that children with more educated mothers in developed counties gain more development benefits than any other group, and the education effect becomes more pronounced in counties with the least GDP compared with those highest GDP ones. Thus, our findings support the double jeopardy hypothesis that low-SES children suffer a double disadvantage to ECD outcomes due to the interactive effects of community socioeconomic characteristics and individual socioeconomic position. An alternative explanation is that people who live in poor areas are limited to access to affordable health care, particularly those disadvantaged children and their families who need it most. By contrast, in wealthy regions, it provides an opportunity to improve the material circumstances of the family and to ensure access to institutional resources and public services, and this benefits children’s early health and growth outcomes and consequently narrow the gaps across socioeconomic levels [78]. Hence, differences in socioeconomic resources across geographic regions result in inequalities in early health development.

As aforementioned, a large volume of studies from western societies has emphasized the role of contextual effects in early child development. However, limited is known in East Asian countries, especially the mediating role of the macro environment in shaping the relationships between family SES and ECD. In this study, we have demonstrated that double jeopardy and concentrated advantage coexist in contemporary China, playing at a county-level to promote or mitigate developmental inequalities from early on. The significant association with geographic socioeconomic contexts and ECD, could be explained by a two-by-two matrix. As shown in Figure 2, the level of local socioeconomic conditions is beneficial not only to improve the outcome for very young children directly, but also to create a more equitable early care and education system to help all children grow up to reach their full potential. Our study helps to extend the existing discussion of the macro environment and ECD in Western countries to broader societies.

These findings also yield a policy implication that policies aimed to improve population health should be intensified in socioeconomically disadvantaged children and their families. The seeds of inequality in adolescence and later life are sewn in early childhood, thereby contributing to the intergenerational transmission of disadvantage. To ultimately eliminate inequalities in health outcomes, special attention needs to be paid to the poorly educated groups, particularly for those in the most deprived areas.

The study has several limitations. First of all, our analyses captured many but not all of the influential factors because of the data limitations. Based on retrospective data, many explanatory variables are not included, such as maternal health status during pregnancy. Moreover, education is the only measurement of socioeconomic resources in this study, and the lack of measures of socioeconomic resources has limited analysis of SES gradients in ECD. Thirdly, we acknowledge that our study fails to address selection bias due to data limitations that may hinder the inference of causal relationships between the macro environment and individual outcomes. In future studies, we will more carefully consider some individual- and family-level variables related to residential self-selection, such as personality traits, or use an instrumental variable approach to reduce or eliminate the risk of selection bias [79]. Finally, our study did not address the potential mechanisms of how the broader social and economic advantage/disadvantage is transmitted into ECD. Future research should attempt to elucidate the interaction between individual factors and the macro environment, in particular for a society like China which is going through major social transformations. Although quantitative data can explore possible trends in large populations, localized and mixed-method studies would be necessary for understanding how family SES, parental expectations, and childrearing practices are differentially affected by county-level variables and socioeconomic status.

## 5. Conclusions

Delays in ECD are of the greatest urgency on a global scale, but special attention needs to be paid to the most considerable number of vulnerable children in developing or under-developed countries. In this study, we estimated the contextual effects of geographic socioeconomic conditions on the achievement of early development milestones in today’s China. Taken together, socioeconomic conditions at the macro-level can shape the ECD in a direct or indirect way as local socioeconomic growth not only promotes children’s developmental health, but also moderates the relationship between individual socioeconomic position and children’s outcomes. Considering multiple dimensions of child development, this study also extends the current research on ECD by revealing the progress toward healthy child development in a range of domains, such as motor, language, cognition, and self-help. Our findings highlight that consistent efforts need to be made to allocate resources and services to the worse-off groups. This also suggests that public health efforts to reduce the intersecting inequalities in ECD should be targeted at regions where the need is greatest. Since China’s birthrate hit a historic low in 2019, it has worsened a looming demographic crisis and an every child matters policy is therefore highly recommended to start from early on and to strengthen preventative services for the most disadvantaged groups in the deprived counties or neighborhoods.

## Figures and Tables

**Figure 1 ijerph-17-04937-f001:**
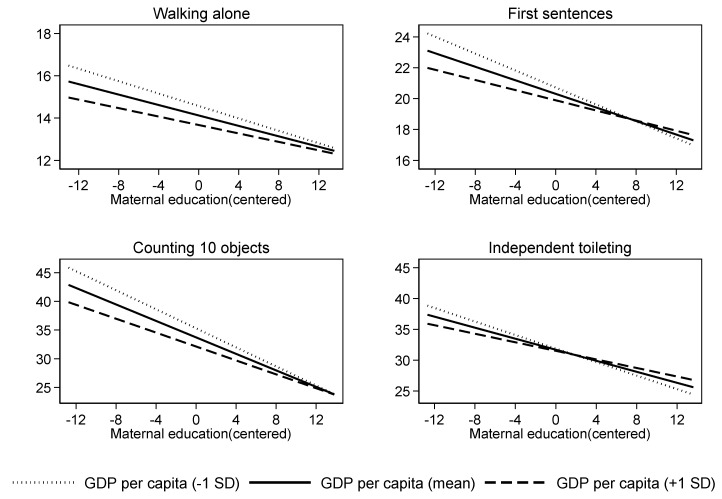
Predicted age of attainment of development milestones with maternal education as a function of county-level GDP per capita.

**Figure 2 ijerph-17-04937-f002:**
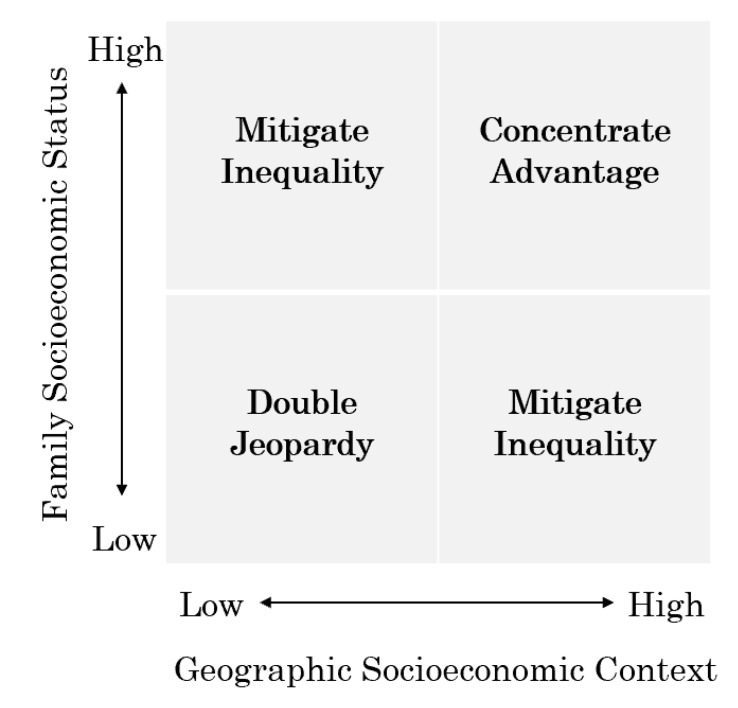
A Conceptual Framework of Intersecting Socioeconomic Gradient and Geographic Context towards ECD.

**Table 1 ijerph-17-04937-t001:** The attainment of developmental milestones by explanatory variables.

Variables	Walk Alone (in Months)		Late Walking		First Sentences (in Months)		Late Talking		Count 10 Objects (in Months)		Late Counting		Independent Toileting (in Months)		Late Toilet-Training	
	Mean	*Sig.*	%	*Sig.*	Mean	*Sig.*	%	*Sig.*	Mean	*Sig.*	%	*Sig.*	mean	*Sig.*	%	*Sig.*
Total	14.21		21.18		20.35		14.11		33.91		23.43		31.66		17.61	
Maternal education		***		***		***		***		***		***		***		***
No schooling	15.87		35.99		22.23		22.60		40.22		39.70		34.50		24.89	
Primary school	14.29		22.38		20.69		14.62		35.98		29.40		32.85		21.23	
Middle school	13.65		16.79		19.54		10.78		31.87		17.52		30.27		13.97	
High school	13.34		12.31		19.32		10.51		29.30		10.02		29.46		12.09	
College	13.12		9.25		18.72		6.98		26.29		4.49		28.95		9.23	
Gender						***		***		**		***				**
Male	14.25		21.56		20.75		15.34		34.54		24.62		32.00		18.08	
Female	14.17		20.74		19.91		12.72		33.41		22.30		31.27		17.07	
Birth cohort		***		***		***		***		***		***		***		
1995–1999	15.25		30.08		21.79		20.52		38.44		36.34		35.56		27.28	
2000–2004	14.66		25.80		20.82		16.21		36.82		32.47		34.68		25.10	
2005–2009	13.79		18.40		19.60		11.80		31.30		15.58		29.28		12.15	
after 2010	13.47		13.59		19.41		8.95		30.12		11.72		28.14		8.48	
Minority status		***		***				^†^		***		***		***		**
Han	14.07		19.66		20.37		13.87		33.77		22.91		31.77		18.21	
Minority	15.31		32.75		20.20		15.93		35.91		28.42		30.81		12.99	
Birth weight		***		***		***		***		***		***		***		***
Low birth weight	15.30		33.28		22.16		19.33		36.42		29.80		32.67		19.50	
Normal weight	13.90		18.23		19.97		12.78		32.67		19.99		30.99		16.11	
Unknown	16.47		41.30		22.61		22.52		44.23		50.45		36.96		29.78	
Gestational age		**		***												
<37 weeks	14.83		29.56		20.48		12.53		33.41		20.05		31.21		16.41	
≥37 weeks	14.19		20.85		20.35		14.17		34.03		23.66		31.67		17.65	
Hukou before age 3		***		***		***		***		***		***		***		***
Rural	14.42		23.13		20.60		15.03		35.29		26.66		32.05		18.83	
Urban	13.31		12.69		19.22		9.92		28.34		9.63		29.90		12.22	
Mother’s age at delivery				**										^†^		^†^
<20	14.52		25.06		20.49		15.19		34.04		25.07		31.30		17.79	
20–24	14.20		21.06		20.20		13.88		33.78		22.63		31.67		17.87	
25–29	14.11		19.70		20.32		13.59		33.86		23.39		31.64		17.09	
30–34	14.32		23.56		20.61		14.60		34.74		25.34		32.16		19.35	
35+	14.44		22.46		20.76		16.74		34.51		24.36		30.62		14.83	
N	11,361				10,324				10,061				10,751			

Notes. † *p* < 0.10, * *p* < 0.05, ** *p* < 0.01, *** *p* < 0.001; *Sig*. denotes significance.

**Table 2 ijerph-17-04937-t002:** Two-level Logistic Regression Models of Early Developmental Milestones on Individual and County-level Variables.

	Walk Alone (in Months)					First Sentences (in Months)					Count 10 Objects (in Months)					Independent Toileting (in Months)				
	(1)	(2)	(3)	(4)	(5)	(1)	(2)	(3)	(4)	(5)	(1)	(2)	(3)	(4)	(5)	(1)	(2)	(3)	(4)	(5)
Individual-level coefficients																				
Maternal education			−0.071 ***	−0.062 ***	−0.302 **			−0.153 ***	−0.144 ***	−0.711 **			−0.442 ***	−0.404 ***	−1.564 ***			−0.222 ***	−0.206 ***	−1.245 ***
			(0.012)	(0.013)	(0.115)			(0.021)	(0.025)	(0.228)			(0.037)	(0.045)	(0.415)			(0.032)	(0.039)	(0.362)
Female			−0.100	−0.103	−0.105			−0.866 ***	−0.871 ***	−0.878 ***			−1.076 ***	−1.087 ***	−1.101 ***			−0.800 ***	−0.790 ***	−0.796 ***
			(0.078)	(0.078)	(0.078)			(0.142)	(0.142)	(0.142)			(0.249)	(0.248)	(0.248)			(0.214)	(0.213)	(0.213)
Birth Cohort (Ref: 1995–1999)																				
2000–2004			−0.391 ***	−0.394 ***	−0.400 ***			−0.644 **	−0.622 **	−0.634 **			−0.437	−0.426	−0.448			−0.297	−0.282	−0.299
			(0.119)	(0.119)	(0.119)			(0.210)	(0.210)	(0.210)			(0.365)	(0.364)	(0.364)			(0.318)	(0.318)	(0.318)
2005–2009			−1.036 ***	−1.024 ***	−1.029 ***			−1.491 ***	−1.480 ***	−1.490 ***			−4.575 ***	−4.597 ***	−4.617 ***			−5.261 ***	−5.274 ***	−5.292 ***
			(0.118)	(0.118)	(0.118)			(0.210)	(0.210)	(0.210)			(0.370)	(0.370)	(0.370)			(0.319)	(0.318)	(0.318)
After 2010			−1.193 ***	−1.191 ***	−1.196 ***			−1.485 ***	−1.482 ***	−1.491 ***			−5.131 ***	−5.116 ***	−5.139 ***			−5.969 ***	−5.979 ***	−5.994 ***
			(0.118)	(0.118)	(0.118)			(0.216)	(0.215)	(0.215)			(0.377)	(0.377)	(0.377)			(0.324)	(0.324)	(0.324)
Minority			0.473 **	0.361*	0.359*			−0.121	−0.160	−0.167			−0.697	−0.575	−0.608			−0.062	0.052	0.042
			(0.168)	(0.167)	(0.167)			(0.316)	(0.315)	(0.315)			(0.559)	(0.560)	(0.559)			(0.479)	(0.472)	(0.471)
Birth Weight (Ref: Normal Weight)																				
Low birth weight			0.769 ***	0.769 ***	0.767 ***			1.640 ***	1.652 ***	1.648 ***			2.103 ***	2.081 ***	2.068 ***			1.540 **	1.617 **	1.611 **
			(0.186)	(0.185)	(0.185)			(0.333)	(0.333)	(0.333)			(0.592)	(0.591)	(0.591)			(0.503)	(0.502)	(0.502)
Unknown			1.309 ***	1.323 ***	1.297 ***			1.255 ***	1.275 ***	1.227 ***			5.683 ***	5.659 ***	5.561 ***			2.499 ***	2.545 ***	2.462 ***
			(0.149)	(0.150)	(0.150)			(0.263)	(0.265)	(0.266)			(0.465)	(0.468)	(0.469)			(0.405)	(0.407)	(0.408)
Gestational Age (Ref: ≥37 Weeks)																				
<37 weeks			0.691 **	0.717 ***	0.713 ***			−0.048	0.008	0.006			−0.370	−0.372	−0.374			−0.221	−0.224	−0.231
			(0.211)	(0.210)	(0.210)			(0.383)	(0.382)	(0.382)			(0.683)	(0.682)	(0.682)			(0.586)	(0.585)	(0.584)
Urban hukou before age 3			−0.296 *	−0.296 *	−0.315 *			−0.334	−0.341	−0.389			−2.484 ***	−2.558 ***	−2.654 ***			−0.565	−0.556	−0.629 †
			(0.127)	(0.127)	(0.127)			(0.237)	(0.236)	(0.237)			(0.413)	(0.413)	(0.414)			(0.353)	(0.352)	(0.352)
Maternal Age at Delivery (Ref: <20)																				
20–24			0.047	0.055	0.052			−0.263	−0.270	−0.275			0.661	0.592	0.583			0.007	−0.014	−0.020
			(0.221)	(0.221)	(0.221)			(0.399)	(0.398)	(0.398)			(0.703)	(0.702)	(0.702)			(0.604)	(0.603)	(0.603)
25–29			0.130	0.139	0.129			−0.069	−0.066	−0.085			1.159	1.113	1.081			0.250	0.235	0.205
			(0.223)	(0.222)	(0.222)			(0.402)	(0.401)	(0.401)			(0.708)	(0.707)	(0.707)			(0.609)	(0.607)	(0.607)
30–34			0.224	0.251	0.245			0.041	0.059	0.050			1.526 *	1.523 *	1.506 *			0.470	0.466	0.451
			(0.235)	(0.234)	(0.234)			(0.424)	(0.423)	(0.423)			(0.748)	(0.747)	(0.746)			(0.641)	(0.640)	(0.639)
35+			0.327	0.357	0.358			0.127	0.175	0.183			1.472 †	1.558 †	1.569 †			−0.663	−0.608	−0.604
			(0.263)	(0.262)	(0.262)			(0.477)	(0.476)	(0.476)			(0.843)	(0.843)	(0.842)			(0.717)	(0.716)	(0.716)
County-Level Coefficients																				
Ln (GDP per capita)		−0.581 ***	−0.453 ***	−0.411 ***	−0.469 ***		−0.549 **	−0.437 *	−0.298 †	−0.436 *		−2.072 ***	−1.594 ***	−1.444 ***	−1.668 ***		−0.402	−0.130	0.109	−0.196
		(0.088)	(0.084)	(0.079)	(0.084)		(0.178)	(0.175)	(0.165)	(0.174)		(0.333)	(0.310)	(0.298)	(0.309)		(0.281)	(0.269)	(0.244)	(0.266)
Urban population %		−0.003	−0.001	−0.000	−0.000		−0.005	−0.001	−0.003	−0.002		−0.068 ***	−0.047 ***	−0.042 ***	−0.040 **		−0.028 *	−0.026 *	−0.025 *	−0.023 *
		(0.004)	(0.004)	(0.003)	(0.003)		(0.007)	(0.007)	(0.007)	(0.007)		(0.013)	(0.013)	(0.012)	(0.012)		(0.011)	(0.011)	(0.010)	(0.010)
Cross-Level Interaction Coefficients																				
Maternal education × Ln (GDP per capita)					0.024 *					0.058 *					0.118 **					0.105 **
					(0.012)					(0.023)					(0.042)					(0.036)
Constant	13.940 ***	19.917 ***	19.054 ***	18.618 ***	19.217 ***	20.120 ***	25.796 ***	25.950 ***	24.574 ***	25.982 ***	32.441 ***	55.737 ***	52.607 ***	50.961 ***	53.270 ***	31.364 ***	36.291 ***	36.709 ***	34.247 ***	37.344 ***
	(0.088)	(0.842)	(0.836)	(0.781)	(0.832)	(0.161)	(1.703)	(1.716)	(1.621)	(1.713)	(0.367)	(3.181)	(3.043)	(2.931)	(3.042)	(0.253)	(2.684)	(2.644)	(2.406)	(2.632)
N	11,361	11,361	11,361	11,361	11,361	10,324	10,324	10,324	10,324	10,324	10,061	10,061	10,061	10,061	10,061	10,751	10,751	10,751	10,751	10,751

Notes. Standard errors in parentheses; † *p* < 0.10, * *p* < 0.05, ** *p* < 0.01, *** *p* < 0.001.

**Table 3 ijerph-17-04937-t003:** Two-level Logistic Regression Models of Early Developmental Milestones on Individual and County-level Variables.

	Late Walking	Late Talking	Late Counting	Late Toilet-Training
Individual-Level Coefficients				
Maternal education	−0.048	−0.218 *	−0.105	−0.239 *
	(0.076)	(0.098)	(0.078)	(0.103)
Female	−0.067	−0.246 ***	−0.156 **	−0.103 †
	(0.049)	(0.060)	(0.052)	(0.055)
Birth cohort (ref: 1995–1999)				
2000–2004	−0.181 **	−0.223 **	−0.049	−0.023
	(0.068)	(0.080)	(0.068)	(0.071)
2005–2009	−0.527 ***	−0.494 ***	−0.900 ***	−0.900 ***
	(0.072)	(0.085)	(0.077)	(0.081)
After 2010	−0.798 ***	−0.739 ***	−1.119 ***	−1.229 ***
	(0.075)	(0.092)	(0.082)	(0.088)
Minority	0.278 **	0.147	−0.124	−0.129
	(0.099)	(0.128)	(0.112)	(0.131)
Birth weight (ref: normal weight)				
Low birth weight	0.530 ***	0.312 *	0.322 **	0.191
	(0.103)	(0.126)	(0.116)	(0.126)
Unknown	0.459 ***	0.221 *	0.573 ***	0.250 **
	(0.080)	(0.097)	(0.083)	(0.091)
Gestational age (ref: ≥37 weeks)				
<37 weeks	0.508 ***	−0.098	−0.188	−0.049
	(0.121)	(0.167)	(0.148)	(0.155)
Urban hukou before age 3	−0.200 *	−0.203 †	−0.605 ***	−0.254 *
	(0.090)	(0.110)	(0.102)	(0.099)
Maternal age at delivery (ref: <20)				
20–24	0.008	−0.128	−0.051	−0.095
	(0.133)	(0.162)	(0.143)	(0.154)
25–29	0.028	−0.156	0.033	−0.106
	(0.135)	(0.163)	(0.144)	(0.155)
30–34	0.201	−0.132	0.079	−0.038
	(0.141)	(0.173)	(0.152)	(0.163)
35+	0.145	0.034	0.109	−0.298
	(0.159)	(0.192)	(0.172)	(0.187)
County-level coefficients				
Ln (GDP per capita)	−0.371 ***	−0.092	−0.299 ***	−0.022
	(0.054)	(0.066)	(0.062)	(0.071)
Urban population %	0.000	0.001	−0.008 **	−0.006 *
	(0.002)	(0.003)	(0.003)	(0.003)
Cross-level interaction coefficients				
Maternal education × Ln (GDP per capita)	0.000	0.017 †	0.004	0.019 †
	(0.008)	(0.010)	(0.008)	(0.010)
Constant	2.425 ***	−0.538	2.305 ***	−0.717
	(0.526)	(0.647)	(0.606)	(0.693)
N	11,361	10,324	10,061	10,751

Notes. Standard errors in parentheses; † *p* < 0.10, * *p* < 0.05, ** *p* < 0.01, *** *p* < 0.001.

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
