# Peer review of "Double Jeopardy in Contemporary China: Intersecting the Socioeconomic Gradient and Geographic Context on Early Childhood Development"

_ijerph, 2020, doi:10.3390/ijerph17144937_

Round 1
Reviewer 1 Report
The topic of this paper is very interesting. However, I don’t think this paper of current condition meets the standard of the International Journal of Environment Research and Public Health for publication due to the following shortcomings:
- Indicators used in this study are not convincing. Using recall data of the parent-reported age of achieved milestones to measure ECD is unreliable. First, it is hard to recall the exact months for the milestones since, especially considering the age of children in CFPS ranges from 0 to 16. Second, social desirability could bias the answers of parents. Besides, the family SES is too complicated to be measured by maternal education unless the authors can provide more evidence/argument.
- The correlation between economic development (measured by GDP per capita in this study) and has been well established.
- The internal validity of this study should be more discussed.
- I suggest the authors show the distribution of children across different waves of CFPS.
- Overall, I do not think the title and abstract are clearly describing what the authors did. I am not sure what do geographical contexts and socioeconomic gradients refer to after reading through the manuscript.
Reviewer 2 Report
The authors of this manuscript looked at how broader socioeconomic context plays a role in young children's passing a set of predictable developmental milestones in China.
Although the study is interesting, I have concerns about the choice of the developmental milestones. The authors have chosen walking, speech, counting and toilet training as developmental milestones. Speech and walking are naturally occurring milestones while toilet training and counting need to be taught. Delays in counting or toilet training usually do not require immediate attention and are also highly cultural as in Western society children usually don't start counting to ten until they are 4 or 5 years of age, much later than in China.
Below are more detailed comments:
Line 15: 'delays in early childhood development is' should be 'delays in early childhood development are important predictors'.
It is a little odd to say that delays are important predictors of success. I think the authors mean delays can cause lack of success or something similar.
17: 'Using a national representative data' should be 'using national representative data'.
52: parenting quality may be undermined in deprived neighborhoods: might be helpful to explain why this would be the case. For example, low levels of education or intergenerational poverty and incomplete education.
59: 'researchers have been increasingly explored' should be 'researchers have increasingly explored' and 'how macro environment interact' should be 'how the macro environment interact'.
62: 'doubly disadvantaged to health' should be 'doubly disadvantaged in health'.
63/64: disadvantaged children face the worst prospects in some affluent areas: this is unclear and needs some explanation. What do the author mean with this?
68: 'very few' should be 'very few studies'.
75/76: 'geographic disparities in today's China is tremendous', should be 'geographic disparities are tremendous.
79: 'Liu et al. (2020) have observed' should be 'Liu et al. (2020) observed'.
83: 'developmental deficits', I would say 'developmental delays'.
104: guardians, could this also be parents?
107: the authors want to compare the Chinese data with Western data but what database are they using for the Western comparison as no mention of it is made in the sample section?
118: instead of 'urinating by oneself' I would recommend toilet trained. Why only urinating and not fully independently toileting? In table 1 on page 5, the authors use both 'urinate by oneself' and 'toilet training'. Why the difference here? Please be consistent and I would recommend using toilet trained.
122: I am wondering why the authors chose counting and urinating on one's own as outcomes as these are milestones that need to be taught and are different from milestones such as walking or smiling which occur naturally? Counting and toilet training are also influenced by cultural norms.
136: please explain what hukou is.
189: 'of the total variation can be explained' should be 'of the total variation and can be explained.
232: On average.....areas: this sentence does not make much sense. I think it should read ' On average, children are more likely to take a first step, speak in sentences, be able to count up to 10 and be toilet trained earlier in more developed areas.
253: what is meant with 'early variables'?
257: 'tackling on early childhood' should be 'tackling early childhood'
274: in 2018, the health expenditure per capita is' should be past tense so 'in 2018, the health expenditure per capita was'.
277: 'and limited to interventions' should be 'with limited access to interventions'
282: 'earn more development benefits' sounds odd and I would rephrase it as 'gain more development benefits'
286-289: this sentence is confusing as it talks about improving benefits in wealthy regions while you would expect that this would apply to children from
The authors state in line 88 and on that they will examine whether the effects follow the same pattern as in Western societies but that is not discussed in this paper, aside from mentioning a couple of reviews but that is different from conducting actual analyses and therefore I would remove this statement from the research question as it is misleading.
Reviewer 3 Report
Thank you for allowing me the opportunity to review this paper titled “Double Jeopardy in Contemporary China: Intersecting the Socioeconomic Gradient and Geographic Context on Early Childhood Development”. The goal of this study is interesting and have potential to make contributions to the field.
The paper follows the standard to the research paper in all parts of the manuscript.
I only have two questions and some suggestions for the authors would be important to clarify before its publication:
- The authors explain in the paper that they capitalize the data from the China Family Panel Studies but they don’t explain if the participants sign an inform consent to participate in this study. Could be important explain how the researchers ask for the consent to the participants, to guarantee the confidentiality and all these ethics issues.
- Habitually, before to the regressions analysis authors present the correlations and the p-value between the study variables. Why authors no present these analyses
- My suggestion is the authors emphasize the idea that in this study they only analyses 2 variables related with the child development. Probably, there are other environment variables such supports that explain the outcome too.
- In the same line, would be beneficial pointed that the results are from oriental country with a particular trait. Probably these findings could be different in different continents.
Congratulations for the study.
Reviewer 4 Report
Review
This study examines how socioeconomic context plays a role in the early development of young children, in both cognitive (speech and counting) and more motor/behaviorial domains (walking and toilet-training) in China. This is an important subject as we need to understand mechanisms behind early childhood development and deprivation better. It has been shown in the literature that early childhood interventions can be quite promising, and ‘repairing’ deficits later in childhood or even in adulthood come at a greater price. This article adds to our knowledge on this, using a rich Chinese dataset. The use of information on age in months for the milestone developments in relation to socioeconomic background helps to magnify the size of the gap and see the geographical differences.
The article is generally well-written and concise, yet sometimes a bit too concise for me. I think a bit more background or explanations can help the reader to put the study and the materials in perspective. Find below my comments that mostly deal with this concern.
Comments and remarks
>> Introduction
My main comment on the introduction is whether the authors can give a bit more information on the exact mechanisms behind the relationship between socio-economic background and deficits in early childhood development. And why they expect the geographical differences? Now things are touched upon, but to readers that are less aware of the literature it is not so clear. Perhaps the authors can also differentiate between theories on the relation between socio-economic background and early childhood development and empirical studies that test the theories at hand.
In l50/53 literature is mentioned about parenting relationships and the authors speak of the possibility that parenting quality may be undermined in deprived neighbourhoods. Are the studies that are cited here of a causal nature? It could well be that the parenting quality would be similar in non-deprived neighbourhoods, with similar effects on childhood development. Care is needed when talking about possible causal inference when analysing the relation with neighbourhood or socio-economic background.
In l64/65 the double jeopardy hypothesis or relative deprivation hypothesis is introduced. I wonder whether the authors can be a bit more specific about this. This is the backbone of their study and I would like to know more about this hypothesis, again also taking account of possible selection effects that I mentioned earlier.
As for the research question, the authors mention that they want to investigate how socioeconomic circumstances affect childhood development. Again, this imposes a causal relation and the question is whether this can be established with the data and hand. Or should the rq’s be rephrased to investigate relationships. This does not make the study of lesser value, but puts more care in the claims that can be made.
>> Materials and Methods
In l105-106 I am a bit confused. The authors mentioned they created a sample of unrepeated individuals. However in the next sentence they talk about children interviewed repeatedly and early they spoke about the longitudinally of the data. Perhaps the authors can be a bit more clear on the dataset and the sampling.
In l112 the author rightly state that children do not develop the milestones in a rigorous timetable. Could they give a bit more background on the distribution of age with respect to the milestones they study? Or some information on the spread at least?
The questions are answered by the parents in retrospect. Do all parents understand the questions in a similar way? Walking alone for example, are any specifics asked about the number of steps? Same applies to speaking brief sentences, what is a brief sentence? Perhaps the authors can tell a bit more on the exact questioning and about possible sensitivity analyses they (or the data constructers) did to ensure these questions are comparable between respondents.
In l132 the authors mention that they use years of schooling as input for the level of education. I am not aware of the Chinese schooling system, but is this comparable across the provinces that are studied? So 7 years of schooling in one province yields a comparable level of education as 7 years of schooling in another?
In l208/211, perhaps it is semantics and interaction effects are generally difficult to explain and interpret, but I guess the positive interaction effect between mother’s education and county GDP means that the positive relation observed between education and development is larger in higher GDP counties. I am not sure whether this is what is stated in these lines. Perhaps the authors can take a look again at the interpretation of the interaction effect.
>> Discussion
One of the main concerns in my mind after reading the results and embarking on the discussion is that I am actually in uncertainy about differences between the counties other than in terms of GDP. So geographic differences are only in GDP. Are there other differences/ variables that the authors might use to give some more background? For example, perhaps some counties are mostly rural and thereby have lower GDP’s. These parents might have more physical intensive jobs, that might also affect development of the children? So it might not necessarily be the lack of health care, but perhaps be related to other factors as well. We do not know whether these parents and children would actually be better off if they were replaced to a more affluent county. This relates again to the mechanisms that I miss in the introduction I guess. I think the study benefits from having somewhat stronger theory in the introduction and a link in the discussion to this. That might also give more body to the conceptual framework they put forward in the discussion. I think this can be quite interesting, but to me needs more body.
In addition, to what extent might these results be generalised to other countries as well? Or are they too typical for China? Perhaps a few sentences on this might increase the value of the study to non-Chinese readers as well.
Round 2
Reviewer 1 Report
I have no more comments or suggestions.
Author Response
Reply to the comments on manuscript Double Jeopardy in Contemporary China: Intersecting the Socioeconomic Gradient and Geographic Context on Early Childhood Development [Manuscript ID ijerph-809548]
Li et al.
July 3, 2020
Reviewer #1:
Dear Mr./Ms. Reviewer:
Enclosed is our newly revised version of Manuscript ID ijerph-809548. Thank you so much for your critical and thoughtful remarks in the first-round major revision, which really helped us to restructure the whole paper and rewrite the key parts. We believe our paper has been significantly strengthened after the first round of revision.
We understand your concerns about further improvement of our research article, in terms of better background information, research design, results outlook and the conclusions. We have worked hard to address all your concerns as follows:
Firstly, we have been more careful in mentioning the related studies as literature review, in terms of the description about the association between the macro environment and the parenting relationships.
Secondary, we made it clearer in the interpretation of results about the Double Jeopardy hypothesis. Our analysis has confirmed that children with less educated mothers in poor areas are facing double disadvantage, that is, limited resources and greater socioeconomic disadvantage associated with child development. Specifically, (1) the county-level socioeconomic background (GDP per capita and the percentage of urban population) is negatively associated with children’s age of achieving early developmental milestones; (2) the county-level socioeconomic background also moderates the relationship between maternal education and early development milestones and reduces the inequality generated by maternal education. In other words, even living in deprived regions, children with better-educated mothers perform better compared to their counterparts whose mother has lower education; furthermore, maternal education matters less in affluent regions and a converging gap between the most- and least-educated groups was observed across regions.
Thirdly, we extended the reflections in research design (i.e., residential selection bias) about the limitations in the conclusion parts.
Finally, we did a thoroughgoing revision of the main text.
Thank you again for reviewing our manuscript in the second-round and providing positive assessments. We appreciate your time and look forward to your response.
Thank you again for all your great kindness during such an unprecedented crisis amidst the COVID-19 pandemic.
Yours sincerely,
Li et al.

Reviewer 4 Report
The authors have revised the manuscript and I appreciate the work they have done. Many things are more clear now. I only have some minor concerns that I wrote down under point three below. I think the manuscript would further benefit if the authors could elaborate a bit more on the points raised there.
Point 1: I can see that the authors are now more clear on that the cited studies are showing observed mechanisms (empirical studies). Before it seemed like there were three theoretical perspectives, but empirical papers were described. I like the restructured paragraph.
Point 2: I see that the authors were a bit more careful in mentioning the studies. I however still wonder whether they should not specifically address the existence of selection bias when studying these relations. Perhaps the authors can do that, mostly in relation to point 3.
Point 3: I appreciate the further explanation of the double jeopardy hypothesis, yet still feel the authors could say a bit more and sometimes I am not sure whether I understand the authors. For example, in this sentence “For example, the protective effect of maternal education on infant diarrhea is larger in the high-poverty areas relative to that in the more affluent communities because basic resources and services is much more constrained by a mother’s knowledge and socioeconomic situation in those poor areas”. Are they saying that children in low ses children are better off in low ses environments? I guess what they would want to say is that within low ses environments, mothers who are a bit higher educated compared to the others in that area are better off, since they understand better how to use the scarce resources? Or am I misunderstanding this?
In addition, the authors mention negative effects of a low ses family in a high ses environment, as part of the second contradicting argument running via stigmatism, as far as I understand. There could also be proof of a third argument, right: the positive effects of a high ses environment on low ses families? If they would be able to be a community member, the could get help and escape the ‘cycle of poverty’.
Point 4: I think the RQ’s are much more in line with the analysis performed and I thank the authors for rewriting.
Point 5: the sampling procedure is now much clearer to me.
Point 6 & 7: I appreciate all the work the authors have done here, as it is much more informative right now.
Point 8: Ok.
Point 9: Ok. I think I understand it now.
Point 10. I appreciate all the work the authors have done here, as it is much more informative right now.
